# TIME TO AUGMENT SELF-SUPERVISED VISUAL REPRESENTATION LEARNING

**Arthur Aubret**[1]* **Markus R. Ernst**[2]* **Céline Teulière**[1] **Jochen Triesch**[2]

[1]Clermont Auvergne Université, CNRS, Clermont Auvergne INP, Institut Pascal
[2]Frankfurt Institute for Advanced Studies
`{arthur.aubret, celine.teuliere}@uca.fr`
`{mernst, triesch}@fias.uni-frankfurt.de`

## ABSTRACT

Biological vision systems are unparalleled in their ability to learn visual representations without supervision. In machine learning, self-supervised learning (SSL) has led to major advances in forming object representations in an unsupervised fashion. Such systems learn representations invariant to augmentation operations over images, like cropping or flipping. In contrast, biological vision systems exploit the temporal structure of the visual experience during natural interactions with objects. This gives access to "augmentations" not commonly used in SSL, like watching the same object from multiple viewpoints or against different backgrounds. Here, we systematically investigate and compare the potential benefits of such time-based augmentations during natural interactions for learning object categories. Our results show that incorporating time-based augmentations achieves large performance gains over state-of-the-art image augmentations. Specifically, our analyses reveal that: 1) 3-D object manipulations drastically improve the learning of object categories; 2) viewing objects against changing backgrounds is important for learning to discard background-related information from the latent representation. Overall, we conclude that time-based augmentations during natural interactions with objects can substantially improve self-supervised learning, narrowing the gap between artificial and biological vision systems.

## 1 INTRODUCTION

Learning object representations without supervision is a grand challenge for artificial vision systems. Recent approaches for visual self-supervised learning (SSL) acquire representations invariant to data-augmentations based on simple image manipulations like crop/resize, blur or color distortion (Grill et al., 2020; Chen et al., 2020). The nature of these augmentations determines what information is retained and what information is discarded, and therefore how useful these augmentations are for particular downstream tasks (Jaiswal et al., 2021; Tsai et al., 2020). Biological vision systems, in contrast, appear to exploit the temporal structure of visual input during interactions with objects for unsupervised representation learning. According to the slowness principle (Wiskott and Sejnowski, 2002; Li and DiCarlo, 2010; Wood and Wood, 2018), biological vision systems strive to discard high-frequency variations (e.g. individual pixel intensities) and retain slowly varying information (e.g. object identity) in their representation. Incorporating this idea into SSL approaches has led to recent time-contrastive learning methods that learn to map inputs occurring close in time onto close-by latent representations (Oord et al., 2018; Schneider et al., 2021). How these systems generalize depends on the temporal structure of their visual input. In particular, visual input arising from embodied interactions with objects may lead to quite different generalizations compared to what is possible with simple image manipulations. For instance, human infants learning about objects interact with them in various ways (Smith et al., 2018). First, infants rotate, bring closer/farther objects while playing with them (Byrge et al., 2014). Second, as they gain mobility, they can move in the environment while holding an object, viewing it in different contexts and against different backgrounds. We refer to (simulations of) such interactions as *natural interactions*.

---

*Equal contribution.

Here, we systematically study the impact of such natural interactions on representations learnt through time-contrastive learning in different settings. We introduce two new simulation environments based on the near-photorealistic simulation platform ThreeDWorld (TDW) (Gan et al., 2021) and combine them with a recent dataset of thousands of 3D object models (Toys4k) (Stojanov et al., 2021). Then we validate our findings on two video datasets of real human object manipulations, ToyBox (Wang et al., 2018) and CORe50 (Lomonaco and Maltoni, 2017).

Our experiments show that adding time-based augmentations to conventional data-augmentations considerably improves category recognition. Furthermore, we show that the benefit of time-based augmentations during natural interactions has two main origins. First, 3-D object rotations boost generalization across object shapes. Second, viewing objects against different backgrounds while moving with them reduces harmful effects of background clutter. We conclude that exploiting natural interactions via time-contrastive learning greatly improves self-supervised visual representation learning.

## 2 RELATED WORK

**Data-augmented self-supervised learning.** The general idea behind most recent approaches for self-supervised learning is that two semantically close/different inputs should be mapped to close/distant points in the learnt representation space. Applying a certain transformation to an image such as flipping it horizontally generates an image that will be very different at the pixel level, but has a similar semantic meaning. A learning objective for SSL will therefore try to ascertain that the representations of an image and its augmented version are close in latent space, while being far from the representations of other unrelated images.

A concrete approach may work as follows: sample an image $x$, apply transformations to it taken from a predefined set of transformations, resulting in new images $x'$, also called a positive pair. The same procedure is applied to a batch of different images. Embeddings of positive pairs are brought together while keeping the embeddings of the batch overall distant from one another. There are three main categories of approaches for doing so: contrastive learning methods (Chen et al., 2020; He et al., 2020) explicitly push away the embeddings of a batch of inputs from one another; distillation-based methods (Grill et al., 2020; Chen and He, 2021) use an asymmetric embedding architecture, allowing the model to discard the "push away" part; entropy maximization methods (Bardes et al., 2022; Ermolov and Sebe, 2020) maintain a high entropy in the embedding space.

**Image manipulations as data-augmentations.** Most self-supervised learning approaches have used augmentations based on simple image manipulations to learn representations. Frequently used are color distortion, cropping/resizing a part of an image, the horizontal flipping of an image, gray scaling the image, and blurring the image (Chen et al., 2020). Other augmentations can be categorized in three ways (Shorten and Khoshgoftaar, 2019; Jaiswal et al., 2021): 1) geometric augmentations include image rotations (Chen et al., 2020) or image translations (Shorten and Khoshgoftaar, 2019); 2) Context-based augmentations include jigsaw puzzle augmentations (Noroozi and Favaro, 2016; Misra and Maaten, 2020), pairing images (Inoue, 2018), greyed stochastic/saliency-based occlusion (Fong and Vedaldi, 2019; Zhong et al., 2020), or automatically modifying the background (Ryali et al., 2021); 3) Color-oriented transformations can be the selection of color channels (Tian et al., 2020) or Gaussian noise (Chen et al., 2020). A related line of work also proposes learning how to generate/select data-augmentations (Cubuk et al., 2019; Tian et al., 2020), but since it takes advantage of labels, the approach is no longer self-supervised.

**Time-based data-augmentations.** Several works have proposed using the temporality of interactions to learn visual representations. A recent line of work proposes to learn embeddings of video frames using the temporal contiguity of frames: Knights et al. (2021) propose a learning objective that makes codes of adjacent frames within a video clip similar, however the system still needs to have information about where each video starts and ends. In contrast, our setups expose the system to a continuous stream of visual inputs. Other methods showed the importance of time-based augmentations based on videos for object tracking (Xu and Wang, 2021), category recognition (Gordon et al., 2020; Parthasarathy et al., 2022; Orhan et al., 2020) or adversarial robustness (Kong and Norcia, 2021). Unlike us they do not make an in-depth analysis of the impact of different kinds of natural interactions. Schneider et al. (2021) showed the importance of natural interactions with objects for learning object

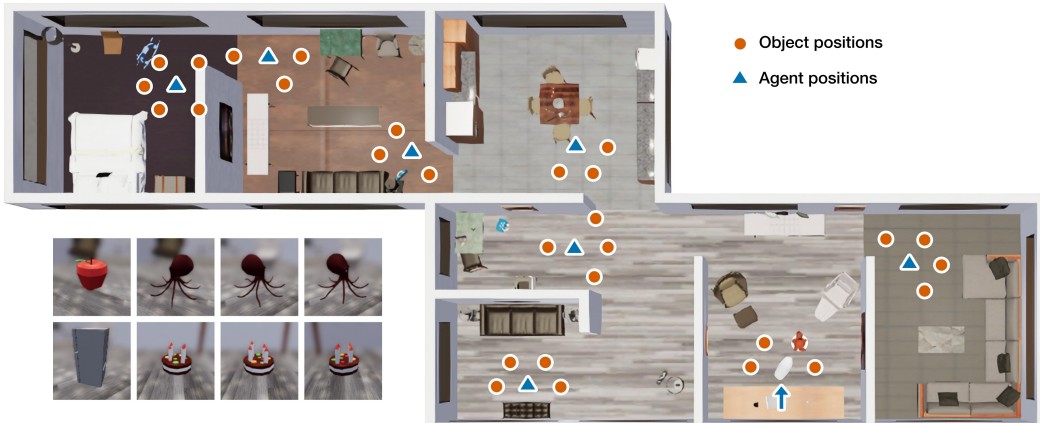

Figure 1: Top view of the VHE used to situate our agent in a house. Blue triangles/orange circles show possible agent/object positions, respectively. In this episode, the agent is located in the office (blue arrow) and interacts with a plush toy. Successive first person views of object interactions are shown in the lower left.

representations, but only considered small datasets of few objects, without complex backgrounds and without studying generalization over categories of objects. Wang et al. (2021) have shown that one can replace crop augmentations by saccade-like magnifications of images, establishing a link between conventional image-based and more natural time-based augmentations. Aubret et al. (2022) studied the impact of embodiment constrains on object representations learnt through time-contrastive learning during natural interactions. They highlight that complex backgrounds negatively impact the representations of objects. Here we show that allowing the agent to move with objects so that they are seen against changing backgrounds improves robustness against background complexity.

We note that a large body of work also considers time-contrastive learning in the context of reinforcement learning (Oord et al., 2018; Laskin et al., 2020; Stooke et al., 2021; Okada and Taniguchi, 2021) and intrinsic motivation (Guo et al., 2021; Yarats et al., 2021; Li et al., 2021; Aubret et al., 2021).

## 3 METHODS

Our goal is to study the potential of time-based augmentations during natural interactions with objects for visual representation learning and to evaluate the utility of the learned representations for object categorization.[1] We focus on two kinds of natural interactions: object manipulations (3D translations and rotations) and ego-motion. We introduce two computer-rendered environments, the *Virtual Home Environment* (VHE) and the *3D Shape Environment* (3DShapeE) to compare and disentangle the effects of these different types of natural interactions.

We also validate our findings using two real-world first-person video datasets. The ToyBox environment (Wang et al., 2018) permits studying the relative benefits of object translations and rotations. The CORe50 environment (Lomonaco and Maltoni, 2017) allows us to explore the effects of smooth visual tracking and to mimic the effects of ego-motion.

### 3.1 VIRTUAL HOME ENVIRONMENT

To simulate a diverse set of natural interactions of an infant in its home, we take inspiration from (Aubret et al., 2022) and place an agent within an environment that resembles a residential house, where it can "play" with objects and change positions. Figure 1 depicts a bird's eye view showing the floor plan of the aforementioned house. In comparison with Aubret et al. (2022), we integrate a two order of magnitude higher number of fully textured toy objects, as well as novel interactions with these objects. The agent is spawned in a randomly sampled location, staying there for 100-timestep-long

---

[1]The source code is available at https://github.com/trieschlab/TimeToAugmentSSL

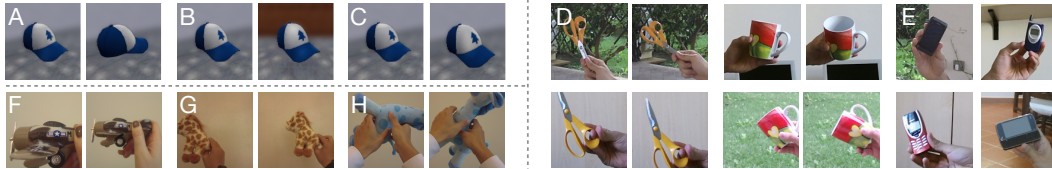

Figure 2: Examples of time-based augmentations while "playing" in the VHE (left, A-C): A) object rotation; B) ego-motion; C) depth change. CORe50 (right, D-E): D) close-by frames of objects in 2 different recording sessions; E) different exemplars of the object category "cellphones" shown in different contexts. ToyBox (left, F-H): F) depth change, G) lateral translation, H) object rotation.

episodes (or sessions). Its position in the world determines what background will be seen. Each background contains a unique combination of static objects and floor/wall textures.

**Toys4k dataset.** The agent interacts with objects from the Toys4k dataset, which is composed of 4,179 diverse 3-D toy objects distributed in an unbalanced way into 105 categories (Stojanov et al., 2021). It was designed to match the objects typically encountered by infants. We place textured objects in different locations of the full house. We modify/remove 3D models, only keeping quick-to-process and correct-quality objects in our simulations (cf. Appendix C.1 for more details). To study the ability of our agent to generalize over categories, we use two thirds of the objects of each category for training and keep the last third for testing. We construct a test set composed of 5 randomly rotated views of the test objects in different locations of the house. Except for rotation, no additional transformations have been applied to build the test set. We also introduce 3 novel houses to test the robustness of the object-representation with respect to novel out-of-distribution backgrounds (cf. Appendix C.2).

**Object manipulations.** At only a few months old, infants start to hold objects and move and rotate them (Byrge et al., 2014). Similarly, our agent can manipulate the objects in two different ways. First, it can rotate the object around the vertical axis (yaw) by a random angle drawn uniformly from the interval $[0, \text{rot}]$ degrees, where the hyperparameter rot sets the maximal speed of object rotations (cf. Figure 2A). Second, the agent can also bring the object closer or move it further away, as shown in Figure 2C; we model this as a change of distance uniformly sampled from $[-d, d]$, where $d = 7.5\,\text{cm}$ is the maximal distance change. The distance is bounded between $0.65\,\text{m}$ and $1.1\,\text{m}$ to reflect morphological limitations of the agent's arms.

**Ego-motion.** When infants become mobile, they can play with objects while moving in their environment. To simulate this, our agent can turn its body towards a neighboring location while holding the same object so that it is seen against a different background. As shown in Figure 2B, the new background will usually have similar floors and walls. Sometimes an infant will also engage with a new object. We model both possibilities by deterministically turning the agent every $N_\text{s}$ time steps and switching to a new object with probability $1 - p_\text{o} = 0.1$. A small/medium $N_\text{s}$ respectively simulates in an unnatural/natural fashion an agent that always/sometimes moves with its object. When this ego-motion is disabled, the agent simultaneously picks up a new object and turns every $N_\text{s} = 10$ time steps. For some experiments (Table 1 and Figure 5A, B), we also allow the agent to visit another room every $N_\text{s}$ steps while continuing to interact with the same object. This allows us to investigate the effect of more drastic background changes, but may be a less valid model of a human infant's interaction with objects.

### 3.2 3D Shape Environment

In order to study the influence of natural interactions on shape generalization while minimizing the impact of object colors or background, we introduce a simple 3D Shape Environment (3DShapeE). It differs from the VHE in two respects: 1) we import untextured versions of toy objects from the Toys4k dataset, making them appear grey; 2) we place the agent in an empty environment so that objects are seen against a blank background. Examples of inputs are given in Figure 4E and augmentations are the same as in the VHE environment (Figure 2A-C).

### 3.3 TOYBOX ENVIRONMENT

The ToyBox dataset (Wang et al., 2018) contains first-person videos of an observer manipulating 360 objects (12 categories, 30 objects each). In each clip, a person applies one type of transformation: a translation (x, y or z), rotation (around x, y or z axis), a "hodgepodge," nothing specific, or in some cases, nothing if there is no object. We sample two frames per second from each clip (cf. Appendix C.3 for more details). Two successively sampled frames form a positive pair. At the end of a clip, a positive pair is formed from the last frame sampled from the clip and the first image of the randomly sampled next clip.

### 3.4 CORE50 ENVIRONMENT

The CORe50 dataset (Lomonaco and Maltoni, 2017) contains first-person views of an observer manipulating objects. The dataset comprises 167,866 images of 50 different objects belonging to 10 distinct object categories. All objects were filmed in 11 sessions corresponding to different natural environments. The images from all but one object per class are used to form the training and validation sets. Specifically, every 10th image enters the validation set, the others form the training set. The images of the held-out object of each class form the test set. This allows us to test generalization to unknown objects from familiar categories.

The videos in this dataset show real object manipulations, but we also use it to mimic the effects of ego-motion through the way we sample images from the different video clips. In general, successively sampled images form positive pairs. We create sequences of images by starting from a particular frame of one of the video clips showing an object manipulation and defining a probability $p_o$ that the manipulation of the same object continues. Otherwise an interaction with a new object will start. In case the interaction continues, the next frame in our sampled sequence will be randomly chosen as either the next or the previous frame in the video clip. Thus, as long as the object manipulation continues, the sampled sequence of frames corresponds to a random walk through the frames of the video. A high/low $p_o$ leads to long/short random walks. Specifically, the expected number of successive frames $N_o$ showing the same object is given by:

$$\mathbb{E}\left[N_o | p_o\right] = 1 + \sum_{x=0}^{\infty} x p_o^x (1 - p_o) = 1 + \frac{p_o}{1 - p_o} \ . \tag{1}$$

To mimic the effects of ego-motion causing an object to be seen against changing backgrounds, we also define a probability $p_s$ that the sequence of views stays within the same session, i.e., the same video clip. Thus, there is a chance of $1 - p_s$ that the sequence of views of the same object "jumps" to a different video clip showing the same object in a different context. In this case, a positive pair is generated from two images of the same object recorded in different recording sessions. This cross session sampling (CSS) is analogous to the role of ego-motion in the VHE.

As an alternative to the random walk procedure described above we also consider a variant where the next view of the same object is chosen uniformly at random from the current video clip. This leads to images forming a positive pair, which stem from distant time points in the original video clip. While this is not a realistic model of viewing sequences experienced by an infant, it allows us to assess the benefits of positive pairs containing more distinct views of the same object. This is analogous to greater speed of object rotation in the VHE.

For implementation details of our sampling procedure see Appendix B.5. In brief, it is an extension of the procedure used in Schneider et al. (2021) to allow CSS.

### 3.5 EVALUATION AND TRAINING PROCEDURE FOR ALL ENVIRONMENTS

**Learning algorithms.** We consider three SOTA representatives of existing algorithms for data-augmented SSL: SimCLR for contrastive learning (Chen et al., 2020), BYOL (Grill et al., 2020) for distillation-based methods and VICReg (Bardes et al., 2022) for entropy maximization methods. We refer to the versions of these algorithms using time-based augmentations as, e.g., SimCLR through time (**SimCLR-TT**), etc. (Schneider et al., 2021).

Time

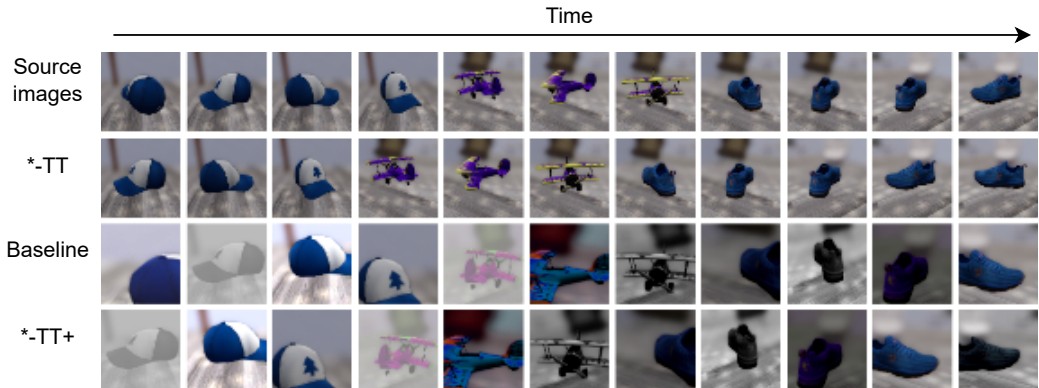

Figure 3: Sequences of input images during object rotations (rot(360)) and ego-motion and their augmented positive pair according to the augmentation method (same column). **Baseline** refers to conventional augmentations; **\*-TT** refers to time-based augmentations; **\*-TT+** refers to combined augmentations.

**Positive pair construction.** We consider three ways to augment an image. First, we consider time-based augmentations during natural interactions (**SimCLR-TT**, etc.). For this, unless stated otherwise, we set the positive pair of a source image as the next image in the temporal sequence. Second, as a comparison baseline, we augment a sampled image with a widely adopted SOTA set of augmentations (Chen et al., 2020; Shijie et al., 2017), *i.e.* crop/resize, grayscale, color jittering and horizontal flip with their default parameters, except for the crop/resize augmentation (cf. Appendix D). Third, we propose to combine the two kinds of augmentations (**SimCLR-TT+**, etc.). For this we apply the conventional SOTA augmentations to the image that temporally follows the sampled source image. We refer to Figure 3 for examples of the different ways to augment source images.

**Controlling for the number of augmentations.** While we can generate an arbitrary number of positive pairs from one sample image with conventional augmentations, we only have access to one positive time-based pair per sample image for the VHE, 3DShapE, and the ToyBox environment. To control for the impact of the number of positive pairs, we therefore only construct one positive pair per sample for all algorithms and augmentation methods: 1) for the ToyBox environment, we sample positive pairs from a pre-built dataset of the same size as the original one, but composed of conventionally augmented images; 2) for the VHE and 3DShapE the agent collects images, augments them and stores them in a buffer of size 100,000. It simultaneously trains on images randomly sampled from the buffer at every time step. We show in Appendix B.1 that controlling for the number of augmentations does not impact the results of our analysis.

**Controlling for the number/distribution of source images.** We want to make sure that we compare the quality of the augmentations, and not the amount/diversity of data that feeds the SSL algorithms. Thus, for the comparison baseline, we consider the same natural interactions to generate source images, but we do not use temporal information to generate the positive pairs. This ensures that the number/distribution of non-augmented images is exactly the same across methods (source images in Figure 3). We make an exception with depth changes in 3DShapE since we found it to be harmful for the baselines.

**Evaluation.** We compute the representation with a ResNet18 on ToyBox and CORe50 and with a simpler convolutional neural network on 3DShapE and VHE. We evaluate the learnt representation after 480,000 steps for the VHE and 3DShapE, 60 epochs for ToyBox and 100 epochs for CORe50. The quality of the learned representation is assessed by training linear readouts on top of the learned representation in a supervised fashion (Chen et al., 2020). Depending on the question, we train linear readouts to either predict the identity of an object, the category of an object, or the location/session of the object. We average all results over 3 random seeds. Because of the small number of objects per category in the CORe50 dataset, we apply a cross-validation with 5 splits. Hyperparameters are given in Appendix D.

| Method | VHE | 3DShapeE | ToyBox | CORe50 |
|---|---|---|---|---|
| SimCLR | $0.328 \pm 0.002$ | $0.527 \pm 0.005$ | $0.313 \pm 0.001$ | $0.544 \pm 0.074$ |
| SimCLR-TT | $0.369 \pm 0.005$ | $0.598 \pm 0.007$ | $0.189 \pm 0.009$ | $0.449 \pm 0.152$ |
| SimCLR-TT+ | $\mathbf{0.537 \pm 0.001}$ | $\mathbf{0.621 \pm 0.001}$ | $\mathbf{0.378 \pm 0.005}$ | $\mathbf{0.610 \pm 0.090}$ |
| BYOL | $0.352 \pm 0.017$ | $0.496 \pm 0.012$ | $0.354 \pm 0.007$ | $0.552 \pm 0.063$ |
| BYOL-TT | $0.369 \pm 0.007$ | $\mathbf{0.570 \pm 0.001}$ | $0.213 \pm 0.014$ | $0.448 \pm 0.160$ |
| BYOL-TT+ | $\mathbf{0.516 \pm 0.002}$ | $0.536 \pm 0.001$ | $\mathbf{0.393 \pm 0.031}$ | $\mathbf{0.614 \pm 0.128}$ |
| VICReg | $0.169 \pm 0.006$ | $0.512 \pm 0.002$ | $0.381 \pm 0.019$ | $0.410 \pm 0.073$ |
| VICReg-TT | $0.264 \pm 0.012$ | $0.383 \pm 0.167$ | $0.158 \pm 0.008$ | $0.245 \pm 0.084$ |
| VICReg-TT+ | $\mathbf{0.435 \pm 0.013}$ | $\mathbf{0.574 \pm 0.008}$ | $\mathbf{0.418 \pm 0.016}$ | $\mathbf{0.503 \pm 0.057}$ |

Table 1: Top 1 accuracy $\pm$ standard deviation (over 3 seeds, CORe50 over 5 training/testing splits) under linear evaluation of previous SOTA (BYOL, SimCLR, VICReg) versus our time-based augmentations (*-TT) and their combination (*-TT+). In 3DShapeE, **\*-TT+** use crop/resize, color jittering and rotations. In VHE, **\*-TT+** use rotations, $N_\mathrm{s} = 1$ with room changes, color jittering and gray scaling. In both VHE and 3D Shape Environments, **\*-TT** use rotations, $N_\mathrm{s} = 1$ with room changes (VHE only) and depth changes. In CORe50 we use $p_\mathrm{s} = 0.5, p_\mathrm{o} = 0.9$. We refer to Appendix B.2 for the ablation study that motivates these choices. In ToyBox, we always apply all their respective transformations. Unlike **\*-TT+**, we apply all standard augmentations for the baseline, as we found that removing some of them (crop/resize and flip) was harmful.

## 4 RESULTS

In this section, we first study the benefits of using time-based augmentations during natural interactions with objects for self-supervised visual representation learning. Then, we provide an in-depth analysis of the impact of 3-D object rotations and ego-motion on the learned representations. An ablation study is performed in Appendix B.

### 4.1 TIME-BASED AUGMENTATIONS DERIVED FROM NATURAL INTERACTIONS BOOST PERFORMANCE OF SSL

In Table 1, we compare the quality of the learnt representation when using conventional data-augmentations (SimCLR, BYOL, VICReg), time-based augmentations (*-TT), and their combination (*-TT+) in our four test environments. We observe that our combined approach (*-TT+) increases the average test category accuracy with respect the the baselines (SimCLR, BYOL, VICReg) by a range of $[0.037; 0.266]$ points across all environments and all SSL methods. To formally validate our approach, we ran additional experiments for SimCLR and SimCLR-TT+ and applied a t-test to statistically compare their performance, supporting the superiority of SimCLR-TT+ ($p < 0.05$ in our four test environments). Time-based augmentations during natural interactions (*-TT) do not perform very well on their own, presumably because they do not create invariance to color/grayscale changes in our test environments (cf. Appendix B for ablation studies of augmentations). In the next sections, we analyse the impact of 3-D rotations and ego-motion on the learnt representations.

### 4.2 OBJECT MANIPULATIONS INCLUDING 3-D ROTATIONS SUPPORT SHAPE GENERALIZATION

To assess the importance of object manipulations to generalize over object shapes, we conducted experiments in 3DShapeE. Figure 4A shows that increasing the speed of object rotations systematically increases the category recognition accuracy. Similarly, in the CORe50 Environment (Figure 4D) the uniform sampling of views clearly outperforms the default random walk sampling procedure, in particular for short object manipulations (low $p_\mathrm{o}$). Interestingly, we observe the inverse effect on individual object recognition accuracy (Figure 4B). We conclude that fast object rotations (in the extreme case constructing positive pairs from randomly rotated views of an object) tend to discard object-specific details and focus the representation on more general shape features, which will be similar for different objects of the same category.

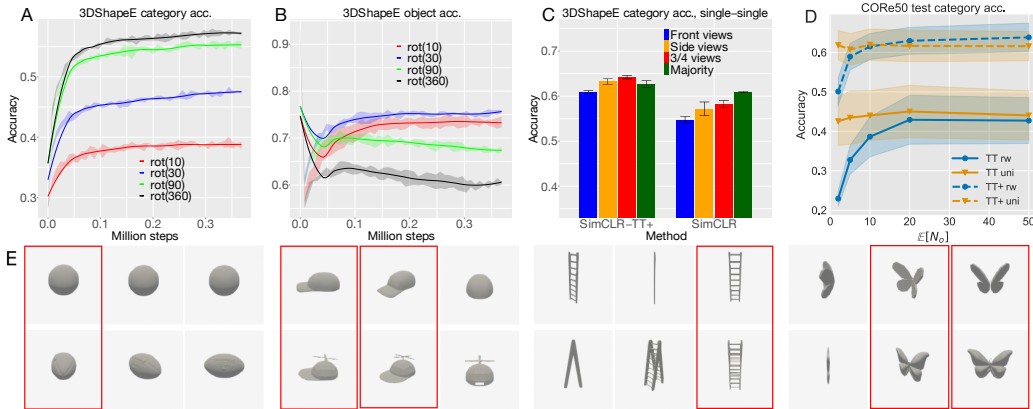

Figure 4: A) 3DShapeE test category recognition accuracy for different maximal rotation speeds $S$, without other augmentations. B) 3DShapeE test object recognition accuracy for different maximum rotation speeds, without other augmentations. The shaded area displays the +/- standard deviation of test category accuracy over seeds. C) Test category accuracy with a linear classifier trained on one single orientation of objects, and tested on the same orientation of test objects. We used 5 seeds, all our conventional augmentations but only rotations ($[0, 360]$ degrees) as time-based augmentation. We also apply the majority vote on datasets of six different orientations: 0°, 10°, -45°, 80°, -80°, -90°. D) Category accuracy for the CORe50 Environment when varying the duration of object manipulations ($p_s = 0.95$). The shaded area shows $\pm$ standard error. A-D) We used SimCLR-TT. E) Example of three views for eight objects from four categories in 3DShapeE. In order from left to right for each object, we show orientations of: 0° (side view); -45° (¾ view); 90° (back view). The red rectangles highlight visually similar views of different objects.

Finally, we investigate the relation between rotation invariance and object shape. To do so, we design a new test setup. In the *single-single setup*, we train the linear classifier on the representations of a single-view dataset that contains only one view from a common orientation for each training object (back, side, …). We assess the classifier on images of the test objects shown in the same orientation. This means that the linear classifier can not learn the invariance to rotation and does not *explicitly* need it for classification. We consider datasets of orientations of -90° (front view), -45° (¾ view), 0° (side view) as they contain the extreme cases in terms of the amount of visible surface (Blanz et al., 1999).

We hypothesize that including 3-D rotations in time-based augmentations allows the model to encode 3-D shape features in the learned representation, which results in an improved category recognition. To test this hypothesis, we first evaluate category recognition based on several views of an object (thus its whole shape). In Figure 4C, we compute a majority vote among classification results from the different single-single classifiers as follows: if more views (e.g., 90°, 80°, 10°) are classified as category A than category B (e.g., 0°, -90°), category A is selected. We observe a large increase of accuracy for SimCLR. We conclude that encoding the 3-D shape improves category recognition. Importantly, we do not observe such an increase of accuracy for SimCLR-TT+ with the majority vote. We deduce that SimCLR-TT+ better infers the 3-D shape of the object from one view, which strongly improves category recognition.

### 4.3 MOVING WITH OBJECTS MITIGATES THE EFFECT OF BACKGROUND CLUTTER

To evaluate the impact of ego-motion while holding objects, we trained our agent in the VHE with different levels of motion. In Figure 5A, we see that increasing the frequency $1/N_s$ and the range (room) of motion considerably boosts the test category accuracy in backgrounds from training and novel houses. At the same time, we observe a decrease of the accuracy when classifying the background (Figure 5B). We conclude that moving while holding objects improves the representation's invariance with respect to the background.

Moreover, we analysed the frequency of session changes compared to the duration of object manipulations as one of the hyperparameters of our approach in the CORe50 Environment, see Figure 5C,

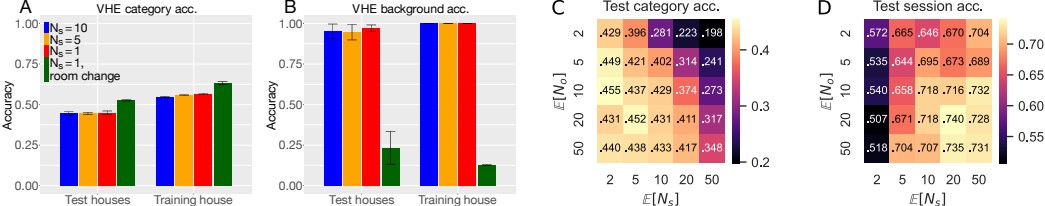

Figure 5: A and B. Impact of the frequency ($N_s$) of body motions and whether they change rooms on the final test category accuracy (A) and the final background accuracy (B). We also used object rotations ($[0, 360]$ degrees), color jittering and gray scaling. Vertical bars indicate one standard deviation. C and D. Mean CORe50 accuracy for category (C) and session (D) classification dependent on the duration of object manipulations ($N_o$) and the frequency of session switches ($N_s$). SimCLR-TT, random walk procedure, mean based on 3 training/testing splits.

D. Qualitatively, for category classification we observe low accuracies for high-frequency object switches, low session changes and high accuracies for low-frequency object switches, high session changes. Accuracy seems to more or less saturate at the main diagonal. We infer from this that intermediate values may actually be better than going to the extreme, but it could also be partly because $N_s$ is bounded by $N_o$. The classification of sessions consistently shows that session information is being disregarded when the frequency of session changes rises.

## 5   CONCLUSION

We investigated the benefits of time-based data-augmentations during infant-inspired natural object interactions for self-supervised visual representation learning. We studied these interactions through a novel 3-D simulation environment where an agent "plays" with objects in a house, and we validated our findings on two real-world video datasets of object manipulations. We find that combining time-based augmentations with conventional data-augmentations (*-TT+ algorithms) greatly improves the ability of the learned representations to generalize over object categories. Our analysis shows that 1) 3-D object rotations are crucial to build good representations of shape categories. 2) Ego-motion while holding an object so that it is seen against different backgrounds prevents the background from cluttering the learned representation.

Our work raises the question whether time-based augmentations during natural interactions with objects could fully replace conventional ones. Indeed, we showed that 3-D rotations are superior to the flip data-augmentation for shape generalization (Appendix B.3). However, we found that color-based transformations like color jittering and grayscaling remain important. It is an interesting question for future work if time-based augmentations during changes of direct and indirect lighting or shadows cast by other objects could replace augmentations based on simple image color manipulations.

Our work has a number of limitations. First, while we have portrayed the time-based augmentations as stemming from object *manipulations*, we did not render a hand holding the object or similar in the VHE and 3DShapeE. Second, the CORe50 and ToyBox environments are only medium-sized. Large scale experiments may give additional insights. Finally, in this work, we only considered naive behavioural strategies for, e.g., turning objects or ego-motion. This contrasts with the way infants learn about their environment (Bambach et al., 2016). For example, they are biased towards creating planar views of objects and often prefer unfamiliar objects (Roder et al., 2000). Thus, we expect that learning to actively select successive views will further unveil the potential of time-based augmentations.

## ACKNOWLEDGEMENTS

This work was sponsored by a public grant overseen by the French National Agency through the IMobS3 Laboratory of Excellence (ANR-10-LABX-0016) and the IDEX-ISITE initiative CAP 20-25 (ANR-16-IDEX-0001). Financial support was also received from Clermont Auvergne Metropole through a French Tech-Clermont Auvergne professorship. We gratefully acknowledge support

from GENCI–IDRIS (Grant 2022-AD011011623R1) for providing computing and data-processing resources needed for this work. Additional support was received by the Deutsche Forschungsgemeinschaft (DFG project "Abstract REpresentations in Neural Architectures (ARENA)"), as well as the projects "The Adaptive Mind" and "The Third Wave of Artificial Intelligence" funded by the Excellence Program of the Hessian Ministry of Higher Education, Science, Research and Art (HMWK). JT was supported by the Johanna Quandt foundation.

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

# A    SAMPLING VIEWS IN THE CORE50 ENVIRONMENT

To simulate $N_o$ and $N_s$ we do not directly sample from the data set, but rather build a buffer of positive pairs and iterate through it.

Building the training buffer is done in cycles, see Figure 6A. In every cycle, each different object of the data set is chosen exactly once — the order in which the objects are presented is a new random permutation during each cycle. For each sampled object, the procedure consists of three steps:

1. We randomly sample a view $v_1$.
2. We sample a view from the same object $v_2$ to form the same-object pair $(v1, v2)$ using a) *random walk*, b) *uniform*. If we reach $N_s$, we sample $v_2$ from a different session (same object but with a different background).
3. We repeat 2. to form $(v_2, v_3)$ and continue until we reached $N_o$ of object manipulation steps.

After that, we sample a view from the next object according to the cycle (which is also a positive pair) and repeat the procedure. Thus, a cycle consists of a total of $N_{obj} \times N_o$ object views. During the manipulation of one object, multiple cross-session transitions can occur, see Figure 6B. Several cycles of $N_{obj} \times N_o$ views are stored in the buffer after another.

During training, batches of pairs of subsequent views are sampled uniformly from the buffer (dashed brackets) and filled into the batch. We consider an epoch to be an entire run through the buffer, which is chosen to match the size of the underlying CORe50 dataset. In consequence, every training image on average is presented once each epoch.

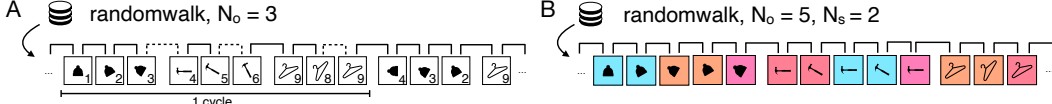

Figure 6: Dynamic sampling procedure for the CORe50 dataset. A) Buffer with deterministic $N_o = 3$. Digits correspond to frame number of the video. B) Buffer with cross session sampling, colors represent different sessions.

# B    COMPLEMENTARY ANALYSIS

## B.1    ADDITIONAL CONTROL EXPERIMENTS

**The number of conventional positive pairs per sample does not matter.**    To fairly compare the quality of time-based augmentations with conventional ones in Table 1, we controlled for the number of conventional positive pairs in VHE, 3DShapeE and ToyBox by fixing 1 positive pair per image. Here, we aim to check whether time-based augmentations are useful even if we increase the number of positive pairs per sample (a new one for each training minibatch). In this case, the number of positive pairs per sample equals the number of epochs.

In the ToyBox environment (Figure 7A), we observe increasing the number of conventional positive pairs improves both the baseline (SimCLR) and the combined augmentations (SimCLR-TT+) and does not change their relative ordering. In addition, it does not significantly impact the accuracy of the 3DShapeE (Figure 7B) and VHE (Figure 7C). Overall, this confirms that time-based augmentations provide complementary and better information about the object shape in comparison to conventional data-augmentations.

**The linear classifier learns about rotation invariance, but it does not fully explain our results.** Here, we want to assess the importance of the role of linear classifiers to learn the invariance over rotations in our results. We focus on the VHE and reuse the *single-single setup* used in Section 4.2 and compare the results with a new *multi-single setup*. Through the *multi-single setup*, we train

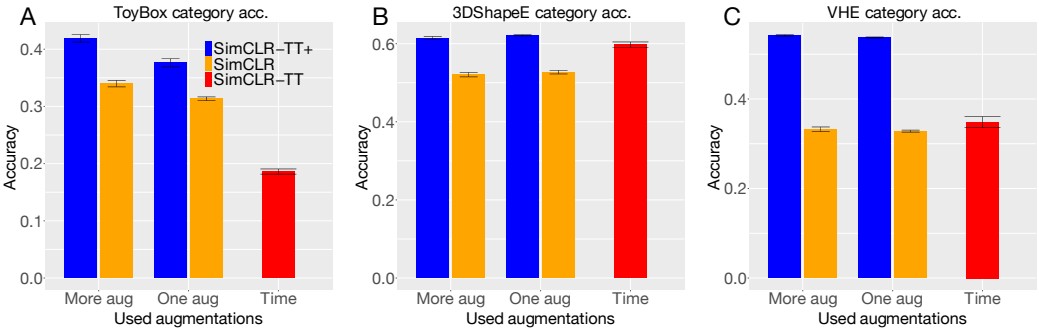

Figure 7: Test category accuracy according to whether we use one positive pair per image (One aug) or several positive pairs, one per minibatch (More aug). We compare on A) the ToyBox environment; B) 3DShapeE and C) VHE.

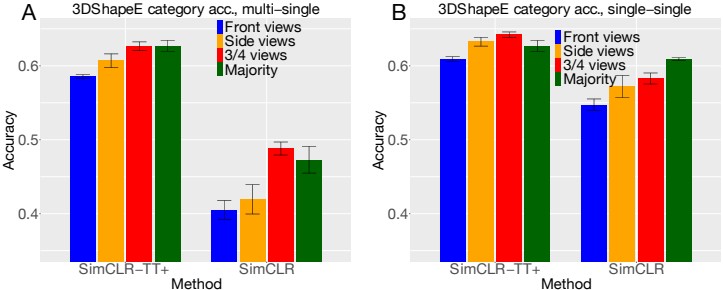

Figure 8: A) Test category accuracy with a linear classifier trained on the multi-view dataset (several orientations of objects), and tested on test objects with one orientation. B) Replicated from Figure 4C for readability. Test category accuracy with a linear classifier trained on one single orientation of objects, and tested on the same orientation of test objects. We used 5 seeds, all our conventional augmentations but only rotations ($[0, 360]$ degrees) as time-based augmentation. We apply the majority vote on datasets of six different orientations: 0°, 10°, 45°, 80°, -80°, -90°.

a linear classifier on representations gained from the usual multi-view dataset, but test it on the single-view datasets.

We observe an overall decrease of performance in SimCLR when testing with the *multi-single setup* (Figure 8A) in comparison to *single-single setup* (Figure 8B). In contrast, SimCLR-TT+ (with rotations only) is more robust to the training set of the linear classifier. It suggests that learning rotation invariance with the linear classifier is mostly harmful for SimCLR. However, the remaining performance discrepancy in the *single-single setup* shows that it does not explain the entire performance discrepancy in Table 1.

**TT+ significantly outperforms its SimCLR counterpart.** As mentioned in the main text we underline the results presented in Section 4.1 by statistically testing our SimCLR-TT+ method against its SimCLR counterpart for all of our 4 different datasets/environments. We did so by rerunning the experiment with additional random-seeds and comparing test-set accuracy with a two sample independent t-test. We found significant differences for all of our comparisons $p < .05$, the details of this analysis can be found in Table 2.

**SimCLR-TT+ also improves over SimCLR in VHE and 3DShapeE with a ResNet18.** We wanted to verify whether *TT+* during object rotations improves over *TT* in VHE and 3DShapeE with a ResNet18 neural network. In Table 3, we repeat the SimCLR-based experiments described in Section 4 with the following modifications: 1– we use a ResNet18 backbone; 2- we set the object manipulation time to 50 time steps; 3- we increase the number of positive pairs per sample (like

| | SimCLR | | SimCLR-TT+ | | | | |
|---|---|---|---|---|---|---|---|
| | $M$ | $SD$ | $M$ | $SD$ | $df$ | $t(df)$ | $p$ |
| VHE | 0.326 | 0.004 | 0.534 | 0.004 | 18 | 112.22 | $\mathbf{2.2 \times 10^{-16}}$ |
| 3DShapE | 0.523 | 0.004 | 0.617 | 0.002 | 17 | 55.477 | $\mathbf{2.2 \times 10^{-16}}$ |
| ToyBox | 0.315 | 0.004 | 0.378 | 0.007 | 10 | 18.409 | $\mathbf{4.8 \times 10^{-09}}$ |
| CORe50 | 0.554 | 0.070 | 0.629 | 0.081 | 18 | −2.228 | **0.039** |

Table 2: Detailed results of the statistical comparison between SimCLR and SimCLR-TT+ regarding test-set accuracy. P-values are bold.

| Environment | SimCLR | SimCLR-TT | SimCLR-TT+ |
|---|---|---|---|
| 3DShapE | $0.617 \pm 0.007$ | $0.598 \pm 0.001$ | $\mathbf{0.674 \pm 0.001}$ |
| VHE | $0.626 \pm 0.002$ | $0.43 \pm 0.007$ | $\mathbf{0.634 \pm 0.003}$ |

Table 3: Results of different augmentations with SimCLR and a ResNet18 on the VHE and 3DShapE environments.

in Appendix B.1); 4- we change the maximum rotation speed in the VHE from 360 degrees to 10 degrees, as we found it to work better in this case. We hypothesize that high rotation speeds favor the encoding of color information, since we did not observe this effect in 3DShapE. We observe that SimCLR-TT+ gives the best performances in both environments.

## B.2 ABLATION OF DATA-AUGMENTATIONS

In this section, we do an ablation experiment of data-augmentations in ToyBox, 3DShapE and VHE since we can choose to remove positive pairs that come from specific natural interactions. We also consider conventional data-augmentations in VHE and 3DShapE since we found the complete set of augmentations to be suboptimal.

**Shape Environment.** In Figure 9A, we test different sets of augmentations in the 3DShapE 3D, progressively and cumulatively combined. Since the combinatorial combination of all augmentations is computationally infeasible, we run a greedy forward algorithm to select the order of augmentations:

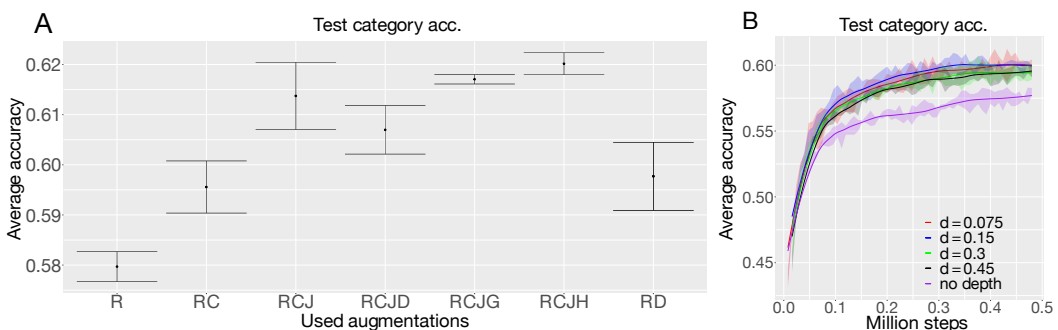

Figure 9: A) Ablation analysis of all data-augmentations in the 3D Shape environment. Horizontal bars indicate the minimum and maximum accuracy over 3 seeds. We note: R=Rotations; D=Depth changes; J=Color Jittering; G=Grayscale; C=Crop and Resize; H=Horizontal flip. B) Analysis of the impact of the maximal change of depth distance $d$.

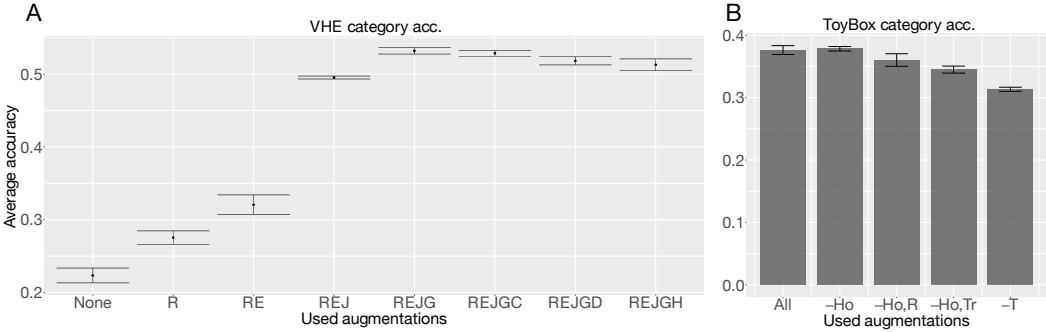

Figure 10: A) Ablation analysis of all data-augmentations in the VHE. Horizontal bars indicate the minimum and maximum accuracy over 3 seeds. We note: R=Rotations; D=Depth changes; J=Color Jittering; G=Grayscale; C=Crop and Resize; H=Horizontal flip; E=Ego-motion with $N_s$=1 and room changes. B) Ablation study on the ToyBox environment. We apply SimCLR-TT+ but remove hodgepodge (-Ho), translations (-Tr), rotations (-R) and all time-based augmentations (-T). Experiments use the same training data, and we only change how we sample the positive pair (except for *None* and Depth experiments).

at each step, we keep the augmentation with the largest increase in accuracy and stop when there is no more improvement. We find three main augmentations: rotations, crop and resize and color jittering. Adding other augmentations does not seem to have a significant large effect on the accuracy.

To better understand the impact of time-based augmentations during depth changes on the representation, we also combine rotations with depth changes (R+D). Adding depth changes achieve similar performance to the addition of crop and resize. We hypothesize it mimics the "resize" part of the crop. An additional analysis (Figure 9B) shows that the results are robust to the maximal speed of depth changes.

**Virtual Home Environment.** We test different sets of augmentations in the VHE, following the same procedure as in Appendix B.2. In Figure 10A, we observe that rotations, ego-motion and color-based transformations are crucial for category recognition. Yet, we can not conclude other augmentations are useless, since it may be the redundancy with other augmentations that is harmful, as we found with depth changes in Figure 9A.

**ToyBox environment.** Figure 10B shows the impact of removing time-based augmentations in the ToyBox environment. We observe that removing augmentations based on translations (-Ho,Tr) and rotations (-Ho,R) both hurt the quality of the representation. Removing only some translations and translations (-Ho) do not affect the performance, presumably because other clips (with rotations and translations alone) already provide information about the shape of the object.

### B.3 ROTATIONS ALLOW FOR BETTER SHAPE GENERALIZATION THAN THE FLIP AUGMENTATION

Here, we investigate how object rotations relate to the standard horizontal flip augmentation, since the flip resembles a 180-degrees rotation for symmetric objects and uniform background. In Figure 11A, we combine the flip augmentation and object rotations with other conventional augmentations. We see that when the object rotation speed is large enough ($rot(360)$), there is no additional benefit in using the horizontal flip augmentation. However, rotations always boost performance compared to the pure flipping augmentation. Thus, we conclude that using 3-D object rotations rather than flipping results in considerably better representation of object shape.

### B.4 HIGH-FREQUENCY OBJECT SWITCHES HURT CATEGORY RECOGNITION.

Here, we study the impact of the object manipulation duration $N_o$. In 3DShapeE (Figure 11B), we observe a sweet spot at $N_o = 10$ when using a simple convolutional neural network (ConvNet). This effect seems to be architecture-dependent since we observe a consistent increase of accuracy as we

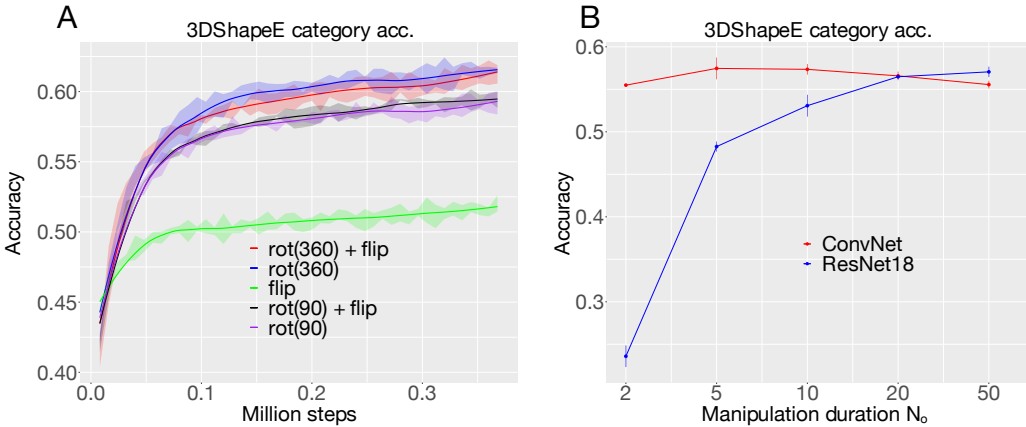

Figure 11: A) 3DShapeE test category accuracy according to different combinations of flip and rotation speeds. B) 3DShapeE test category accuracy according to different manipulation duration $N_o$ with a simple ConvNet and a ResNet18.

increase $N_o$ with a ResNet18. Importantly, in both cases, we observe that high-frequency object switches (low $N_o$) hurt the downstream object categorization.

In CORe50 (Table 4), the combined randomwalk (TT+ rw) method is sensitive to a high frequency of object switches within the same session, which considerably improves validation set category accuracy (A). We assume that this is because the additional contrastive information is considerably higher for the randomwalk method, whereas uniform view sampling already achieves some degree of background variation due to sampling contrasts from one whole session compared to the neighboring frames. For test set accuracies, we observe a similar trend, albeit not as pronounced (B). Note how test-accuracies for the random walk procedure are considerably higher than for the uniform sampling.

### B.5  INFLUENCE OF THE SESSION SWITCH FREQUENCY IN THE CORE50 ENVIRONMENT

**Detailed analysis of varying the session frequency in the CORe50 environment.** To complete the results from the CORe50 dataset we add overview tables and a more thorough analysis regarding the randomwalk procedure. The mean and standard errors of the analysis corresponding to Figure 4D can be found in Table 4 together with a supervised and SimCLR baseline trained with the same hyperparameters. The uniform view sampling (uni) performs best on the validation set, whereas the two combined methods (*-TT+) deliver the best generalization capabilities across splits.

| algorithm | $\mathbb{E}[N_o]$ | | | | |
| --- | --- | --- | --- | --- | --- |
| | 2 | 5 | 10 | 20 | 50 |
| supervised | $.530 \pm .080$ | —— " —— | —— " —— | —— " —— | —— " —— |
| SimCLR | $.544 \pm .030$ | —— " —— | —— " —— | —— " —— | —— " —— |
| -TT rw | $.229 \pm .018$ | $.327 \pm .033$ | $.386 \pm .046$ | $.429 \pm .055$ | $.427 \pm .052$ |
| -TT uni | $.426 \pm .055$ | $.435 \pm .060$ | $.440 \pm .056$ | $.450 \pm .059$ | $.440 \pm .056$ |
| -TT+ rw | $.501 \pm .032$ | $.589 \pm .030$ | $.615 \pm .029$ | $\mathbf{.629 \pm .031}$ | $\mathbf{.638 \pm .032}$ |
| -TT+ uni | $\mathbf{.618 \pm .084}$ | $\mathbf{.608 \pm .036}$ | $\mathbf{.620 \pm .036}$ | $.616 \pm .036$ | $.616 \pm .033$ |

Table 4: Category accuracy and standard error on CORe50, based on SimCLR, $p_s = 0.95$. Standard error based on five different training/testing splits. Training occurred for 100 epochs, batchsize 512. Best performance is highlighted in bold, cf. Figure 4D.

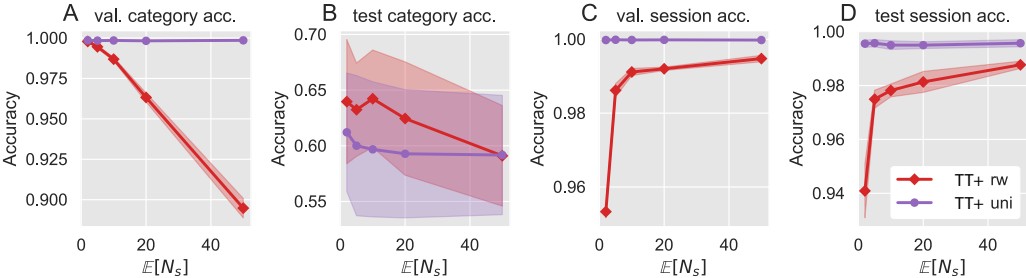

Figure 12: Analysis of the SimCLR-TT+ approach for different values of $p_\mathrm{s}$, CORe50 Environment. Error bars indicate standard error based on three different training/testing splits.

We also report means and standard errors for category and session accuracy (Table 5) of our randomwalk procedure when evaluated with different values for $p_\mathrm{s}$ and $p_\mathrm{o}$.

| | Category accuracy | | | | |
|---|---|---|---|---|---|
| | $\mathbb{E}\left[N_\mathrm{o}\right]$ | | | | |
| $\mathbb{E}\left[N_\mathrm{s}\right]$ | 2 | 5 | 10 | 20 | 50 |
| 2 | $.993 \pm .001$ | $.994 \pm .001$ | $.994 \pm .001$ | $.997 \pm .000$ | $.998 \pm .000$ |
| 5 | $.898 \pm .009$ | $.974 \pm .002$ | $.986 \pm .000$ | $.989 \pm .001$ | $.992 \pm .001$ |
| 10 | $.572 \pm .005$ | $.909 \pm .005$ | $.956 \pm .004$ | $.972 \pm .002$ | $.981 \pm .003$ |
| 20 | $.439 \pm .004$ | $.629 \pm .008$ | $.814 \pm .010$ | $.908 \pm .009$ | $.939 \pm .004$ |
| 50 | $.401 \pm .007$ | $.454 \pm .006$ | $.522 \pm .007$ | $.617 \pm .003$ | $.711 \pm .006$ |

| | Session accuracy | | | | |
|---|---|---|---|---|---|
| | $\mathbb{E}\left[N_\mathrm{o}\right]$ | | | | |
| $\mathbb{E}\left[N_\mathrm{s}\right]$ | 2 | 5 | 10 | 20 | 50 |
| 2 | $.572 \pm .010$ | $.535 \pm .011$ | $.540 \pm .015$ | $.507 \pm .008$ | $.518 \pm .010$ |
| 5 | $.665 \pm .012$ | $.644 \pm .014$ | $.658 \pm .018$ | $.671 \pm .011$ | $.704 \pm .011$ |
| 10 | $.646 \pm .016$ | $.695 \pm .019$ | $.718 \pm .016$ | $.718 \pm .014$ | $.707 \pm .009$ |
| 20 | $.670 \pm .026$ | $.673 \pm .003$ | $.716 \pm .021$ | $.740 \pm .013$ | $.735 \pm .013$ |
| 50 | $.704 \pm .012$ | $.689 \pm .025$ | $.732 \pm .012$ | $.728 \pm .015$ | $.731 \pm .020$ |

Table 5: Category accuracy and standard error on CORe50, SimCLR-TT algorithm with randomwalk sampling procedure. Standard error based on three different training/testing splits. Training occurred for 100 epochs, batchsize 512, cf. Figure 5C, D.

**Influence of session changes on combined augmentations.** Since our main analysis with the CORe50 dataset aims to give a solid overview over the presented approaches, we have picked a medium level of session changes as the default ($p_\mathrm{s} = 0.95$, $\mathbb{E}\left[N_\mathrm{s}|p_\mathrm{s}\right] = 20$). The results show that the combined approaches (*-TT+) outperform the conventional SimCLR method, but we were also interested in how sensitive the novel methods are to the $p_\mathrm{s}$ parameter. Results of that analysis are depicted in Figure 12. In line with results from our Virtual Home Environment experiments, we observe that for very low values of $N_\mathrm{s}$ the internal representation contains less information about the session for the randomwalk approach (C,D). The uniform method, however, shows robustness to this effect.

# C ASSETS

## C.1 3D OBJECTS IN THE VIRTUAL HOME ENVIRONMENT

We import two versions of objects into the ThreeDWorld Software (TDW):

**3DShapeE.** This is the grey version of objects. We export the 3D models to *.obj* format and remove the texture files. We discard objects that took too long to process in TDW.

**Textured objects.** This is the colored version of objects. In order to import them in TDW, we apply a series of operators: 1) we reduce the complexity of some meshes to make the next step computationally feasible and reduce the TDW processing complexity; 2) we manually *bake* most of the 3D models to obtain a single texture file with the meshes; 3) we import the models into TDW; 4) we resize the models to marginalize the effect of size on category recognition and avoid harmful collisions. We visually check the quality of all models. When we do not manage to obtain good-looking textures for some objects, we remove them from our dataset ($< 15\%$ of the dataset). The *tree* category does not contain enough good-looking objects, so we remove the whole category.

## C.2 TEST DATASETS OF THE VIRTUAL HOME ENVIRONMENT.

Our House test set is composed of 7,740 images, from 1,548 objects distributed into 105 categories. By showcasing the category "elephant" in Figure 13 we exhibit the diversity of objects within one category. The elephant category includes flying elephants, bipedal elephants, red elephants, differently positioned elephants or geometrically simplified elephants. We also show the sequence of input images in the VHE according to different rotation speeds.

In Figure 14, we display the three test houses used in Figure 5B,C. House scenes were taken from pre-provided bundles in TDW and we added the agent and objects in the scene similarly to Figure 1.

The underlying ThreeDWorld (TDW) software used is licenced under BSD 2-Clause.

## C.3 TOYBOX ENVIRONMENT

The ToyBox environment is a collection of video-clips. We sample two frames per second from each clip, resulting in 159,627 images. Converted clips contain 2 to 50 sampled frames, which corresponds to the $N_o$ parameter in the CORe50 environment. Two thirds of objects are used for the training set and the rest for the test set.

# D HYPERPARAMETERS

For all conducted experiments, we apply a weight decay of $10^{-6}$ and update weights with the AdamW optimizer (Loshchilov and Hutter, 2018) and a learning rate of $5 \cdot 10^{-4}$.

**Virtual Home Environment and 3D Shape Environment.** The agent in the Virtual Home Environment perceives the world around it as $128 \times 128$ pixel RGB images. Unless stated otherwise, these images are encoded by a succession of convolutional layers with the following [channels, kernel size, stride, padding] structure: [64, 8, 4, 2], [128, 4, 2, 1], [256, 4, 2, 1], [256, 4, 2, 1]. The output is grouped by an average pooling layer and a linear layer ending with $128$ units. Each convolution layer is followed by a non-linear ReLU activation function and a dropout layer ($p = 0.5$) to prevent over-fitting. We do not use projection heads. We consider a temperature hyperparameter of $0.1$ for SimCLR. We use a batch size of 256 and a buffer size of 100,000. Average training time was 72 hours per run for VHE, and 20 hours for the 3DShapeE. All experiments ran on GPUs of type NVIDIA V100.

We did a hyperparameter search of the best crop minimal size in $\{0.08, 0.2, 0.5, 0.75\}$ for the *Combined* and *SimCLR aug* experiments.

**CORe50 Dataset.** The additional diversity of the CORe50 training data required a more capable encoder network, which we chose to be a ResNet-18 (He et al., 2016). The encoder transforms the

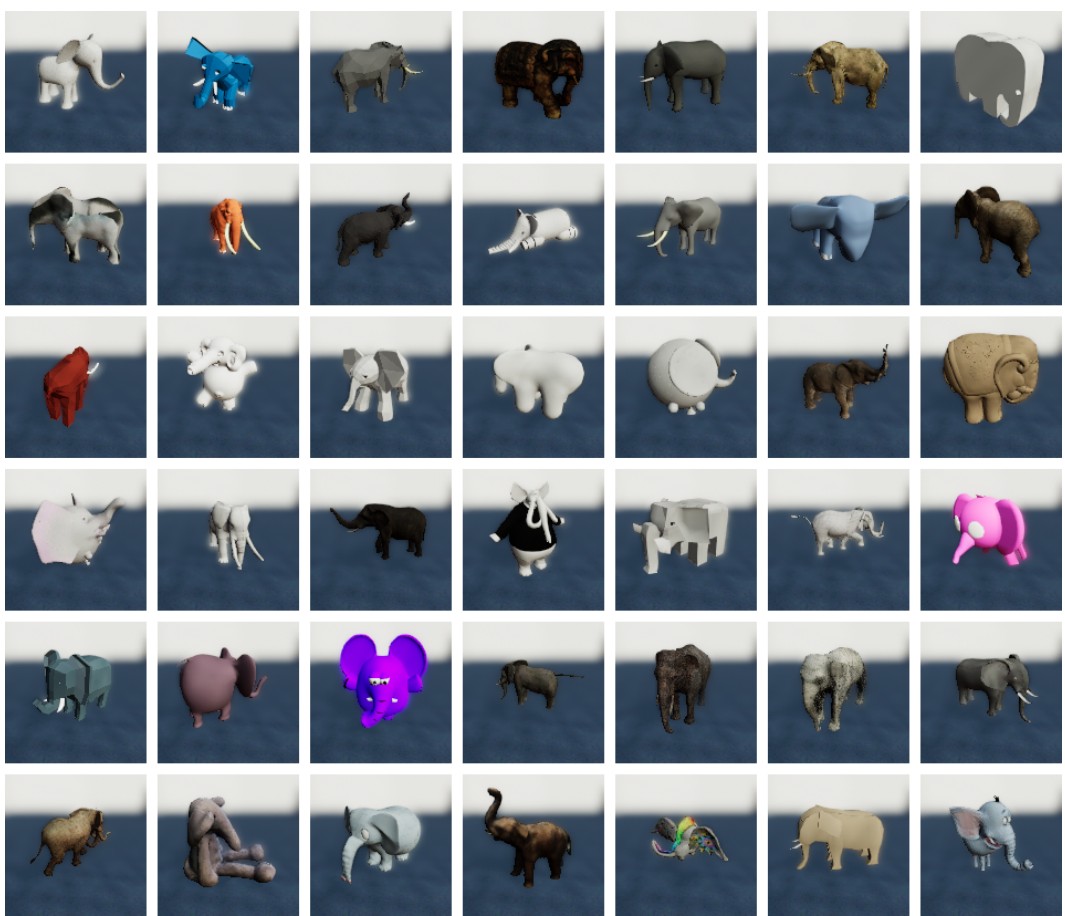

Figure 13: All objects of the category "elephant" used in our Virtual Home Environment.

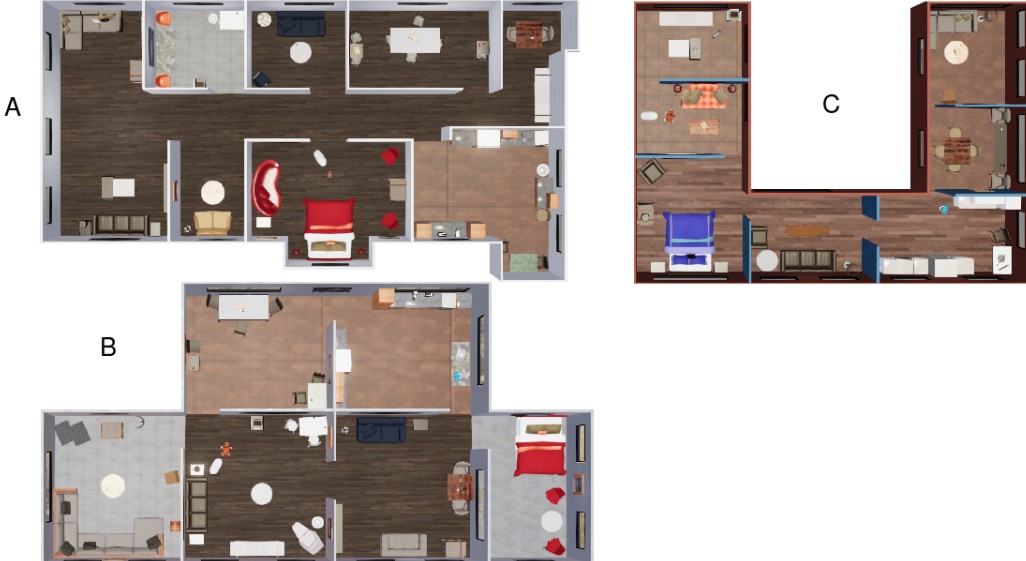

Figure 14: Top view of houses, without ceiling, used only when testing category recognition in unfamiliar houses experiments. A) House 2; B) House 3; C) House 4.

input images into a 128-dimensional latent representation, which is followed by a 2-layer linear projection head [256, 128]. For the projection head, we use batch normalization and a ReLU activation function in-between the two layers. We chose a batch size of 512 and training was done for 100 epochs on a GPU of type NVIDIA V100 or NVIDIA RTX2070 SUPER. Average training time was 17.5h per run. For BYOL we use $\tau = 0.996$ and for VICReg the default error weighting as described in Bardes et al. (2022). The CORe50 dataset is licensed under CC-BY-4.0.

**ToyBox environment.** We use the same hyperparameters as for the CORe-50 experiments. However, we made an ablation study of the crop minimal size hyperparameter to adapt to the small size and off-center position of the objects. We found a value of $0.5$ to be the best in $\{0.08, 0.2, 0.5, \text{None}\}$.

