# OpenReview forum: "Time to augment self-supervised visual representation learning"
_ICLR.cc/2023/Conference — ICLR 2023 poster_

### Official Review · Reviewer_S6Vj · 2022-10-22

**Confidence:** 3
**Correctness:** 3
**Technical Novelty And Significance:** 2
**Empirical Novelty And Significance:** 2
**Recommendation:** 6

**Clarity, Quality, Novelty And Reproducibility:**

- This paper is well written and easy to follow.
- The authors are well thought on the experiments to support their claim, making it a good quality paper
- The augmentations introduced in this paper are not significantly novel perse, but the careful design of the augmentations to mimic human response is likely to be novel.
- If the authors would release their data generation pipeline, this paper is reproducible.


**Strength And Weaknesses:**

#### Strength

This paper excels at performing well designed and controlled augmentations to study their performance in various settings. Through this design and careful control, it is convincing to show that the type of augmentation introduced are helpful for SSL.


#### Weaknesses

My main concern for this paper is the notion of "time-based" augmentations. I find this term a bit misleading: most of the augmentation introduced seems time-invariant, but more as 3D augmentations. By time-invariant I mean in the case that if all the frames captured each session are randomly shuffled then piped to the SSL frameworks, augmentations like manipulation and ego motion seems unaffected at all, where all that matters are the set of positive and negative pairs, but not their temporal relations. Next frame positive pair might be affected, which is truly time-based. But according to [1], this may not be a good strategy for mining semantic correspondences.

I therefore would like to ask the author for more arguments or evidence to support the use of "time", in addition to the 3D-based augmentations provided in the paper. Or a more proper name should be used.

[1] Rethinking Self-supervised Correspondence Learning: A Video Frame-level Similarity Perspective

**Summary Of The Paper:**

This paper studies how augmentations induced by object interaction sessions help with self-supervised learning performance. This paper uses two synthetic environments and two real world data environments, each with different manipulation, ego-motion, and fixation related augmentations for self-supervised learning. The authors showed that these types of augmentations are more helpful than traditional image-space augmentations, and best if combined with those augmentations as well.



**Summary Of The Review:**

In summary, I find this paper slightly below the bar of acceptance. My major concern is the notion of 'time-based augmentations', where a lot of those augmentations are just 3-D based and are time ignorant. To match this term more closely, the augmentations, or the positive frame selection must be time sensitive. To support the claim that this is helpful, the baseline should be a time-ignorant positive frame sampling strategy with the same set of augmentations.
Also, the types of augmentations shown here are marginally novel in pure SSL sense, but this could be an underestimate since the augmentations are designed more closely to mimic human responses.

## Update after the rebuttal
My main concern is mostly around the term 'time-based': some of the augmentations used in this paper is not quintessential in terms of 'time'. One example is rotation of the object, which can be seen as an instance-centric 3D augmentation, where the ordering of different poses along the temporal dimension does not matter.
The longer phrase "time-based augmentations during natural interactions" is more accurate, and I apprecitate the change proposed by the authors. I do hope the title of the paper can be more accurate as well, but it is a relative minor issue.
Given the author's response, I would like to increase my rating from 5 to 6.

---

> ### Author Response · Authors · 2022-11-16
> **Response to review**
>
> Your concern seems to be mostly about terminology and may be rooted in a misunderstanding of how positive pairs are constructed in our approach. We use the term “time-based augmentation” because time (nothing more and nothing less) drives how positive pairs are constructed. Of course, something must “happen” during this time for the approach to work. We show that what happens during simulated and real “natural interactions” with objects leads to much improved representations. To improve clarity, we now use the longer phrase “time-based augmentations during natural interactions” in most places.
>
> If the order of frames was shuffled randomly in our approach, then nothing useful could be learned, because positive pairs would be composed of different views of unrelated objects. Recall that we learn from an unsegmented stream of views that is not grouped into segments corresponding to interactions with the same object (although some of the datasets we use would permit doing so).
>
> Furthermore, our results are actually in line with [1] since we found that the uniform sampling procedure of the CORe50 dataset works better than the random walk procedure for short manipulation duration (cf. Table 3 in Appendix B6). This procedure is conceptually close to the “distant” procedure in [1]. This is also consistent with our observation that large object rotation speeds (corresponding to large view point changes across successive images) are beneficial for category recognition (Figure 3).
>
> Because of the above, we think it is quite appropriate to speak of “time-based augmentations during natural interactions” and we would like to ask you to reconsider your score.

---

### Official Review · Reviewer_Qi6u · 2022-10-23

**Confidence:** 4
**Correctness:** 3
**Technical Novelty And Significance:** 3
**Empirical Novelty And Significance:** 2
**Recommendation:** 6

**Clarity, Quality, Novelty And Reproducibility:**

The overall motivation and introduction are very clear and compelling. Although several aspects of the work are creative and novel, unfortunately the experiments and results do not really live up to the expectations.

I expect reproducibility to be limited, since the relatively large number of individual experimental manipulations are not described in much detail, albeit these things could be inferred from the code, which is promised to be published. *[note: in my original review, I had not seen the note about code being published]*


**Strength And Weaknesses:**

### Strengths

 + Well motivated overall question
 + Important question how active vision could support visual representation learning beyond the simple data augmentation strategies currently used
 + Using rendering could be an interesting way forward to generate realistic view changes synthetically
 + Implemented an interesting and diverse set of view changes caused by active vision not captured by data augmentation


### Weaknesses

 1. Method does not actually result consistently in improved visual representations
 1. Small size of datasets seems to be a major limiting factor preventing strong conclusions
 1. Motivation for the individual experiments and how they each relate to the main research question is not very clear
 1. Analyses are lacking depth



### 1. No consistent improvement

One major concern is that the results do not really show clear evidence that the proposed approach indeed outperforms standard data augmentations. In more than half of the dataset/learning method combinations it actually performs worse.


### 2. Small datasets

Another major concern is what we can actually learn about self-supervised learning in practically relevant scenarios from such small datasets. The fact that both methods (standard and time-based) combined work significantly better than each one individually suggests that in both cases the number of unique image sources/scenes is a major limiting factor and the combination of both methods simply leads to substantially more "effective" data. How each of the methods would fare in a more typical self-supervised setting where the datasets are large is not really answered by these experiments.


### 3/4. Motivation + lack of depth

The authors present a collection of results, each of which addresses a question that is in principle interesting. However, it was not very clear to me why exactly these questions were asked and what we learn from the analyses. To name just one concrete example, why are saccades only simulated on CORe50 but not VHE?

Each question is addressed very superficially by one manipulation, often on a single dataset, without discussing robustness and generality of these results: how much do they depend on the particular way the authors operationalized the question, what about different datasets or change in hyperparameters?

Overall, after reading the paper I'm left wondering: what now? What follows for self-supervised learning? The fact that invariance to 3d rotations and changes in background are useful is undoubted, but such changes simply cannot be easily generated from unlabeled images and deriving them from unlabeled video is not a unique contribution of this paper.


**Summary Of The Paper:**

The authors propose to use temporal changes in object views that occur naturally in developing infants' visual environment as data augmentations for time-based self-supervised learning. Using a rendering environment they create positive and negative training pairs using 3d object manipulations such as rotations, saccadic eye movements, moving objects (to change the background). They compare to standard self-supervised learning using data augmentation and evaluate the learned representations on both the rendered and some real-world video datasets of humans manipulation objects.


**Summary Of The Review:**

A well-motivated paper with creative and original aspects but relatively poor execution and somewhat unclear conclusions.

### [Update after rebuttal]

Based on the discussions and the other reviewers' impressions I increased my score and gave the paper the benefit of the doubt.

---

> ### Author Response · Authors · 2022-11-16
> **Response to review**
>
> - Method does not actually result consistently in improved visual representations
>
> There is clearly a misunderstanding here. In Table 1, we show that combining time-based augmentations with conventional augmentations (*-TT+) consistently outperforms conventional augmentations alone (*). This holds true for four environments/datasets with three different baselines and is our main result.
>
> Perhaps you are referring to the fact that time-based augmentations do not perform well on their own. We have investigated this gap and in Figure 11 we show that the absence of color invariance plays a large role in this. In addition, combining our time-based augmentations during natural interactions with only color-based augmentations (color jitter, gray scaling) achieves an average accuracy of 0.375 (vs 0.339 for SimCLR, Figure 7-A) and 0.563 (vs 0.544 for SimCLR, Table 1) for the ToyBox and CORe50 environments, respectively.
>
> - Small size of datasets seems to be a major limiting factor preventing strong conclusions
>
> We find this statement rather subjective. In fact, all our datasets are two to three times larger than the Cifar10 dataset, which shows meaningful results in SSL (Chen et al. 2020, appendix B9). To address any remaining concern, we will add statistical tests that demonstrate statistical significance of the observed performance improvements before the end of the rebuttal.
>
> - The fact that both methods (standard and time-based) combined work significantly better than each one individually suggests that in both cases the number of unique image sources/scenes is a major limiting factor
>
> We do not understand this argument. First, Combined and SimCLR methods access the exact same number/distribution of non-augmented images. Second, In Table 1, we control for the number of positive pairs per sample (cf. Section 3.5). Not controlling for it does not change our results (Figure 7). We interpret the findings as showing that standard image based augmentations and our time-based augmentations during natural interactions are in fact complementary. This is supported by the finding that time-based augmentations alone do not perform very well, but in combination with color-based augmentations alone they outperform the standard augmentations of SimCLR (compare our reply to your first point).
>
> - Motivation for the individual experiments and how they each relate to the main research question is not very clear
>
> Thank you for this important point. We have updated the text to better motivate the rationale for each experiment: 1) Object manipulations (rotations and translations) are studied with VHE, 3DShapeE, CORe50 and ToyBox; 2) Ego-motion is studied in VHE and the CORe50 environment.
> We note that our results and text about gaze control were confusing (also for Reviewer 1), while not being an important contribution of the paper. Thus, we thus decided to remove them from the paper.  We also replaced the terminology “number of fixations” with “object manipulation duration” since we believe it was a source of confusion: we never studied saccades in the CORe50 dataset.
>
> - Analyses are lacking depth
>
> We have performed additional analyses whose results have entered into the Appendix: In Figure 9B, to clarify our choice of speed of depth change in VHE, we test different maximal speeds of depth change. Furthermore, we will add before the end of the rebuttal a study of the impact of the duration of object manipulations in 3DShapeE. We will also add the results for larger neural network architectures as soon as possible.
>
> - Lack of perspectives
>
> Indeed, there may not be much near-term impact for conventional computer vision on existing standard image databases. But we clearly demonstrate how much improvement time based augmentations based on natural interactions can bring.
>
> - No code provided
>
> As indicated in the manuscript, we are going to publish the code and all resources.

---

> > ### Comment · Reviewer_Qi6u · 2022-11-18
> > **Still wondering**
> >
> > Thanks for clarifying!
> >
> > I do understand that it's the combination that performs best. That's also what worries me, in combination with the datasets being small (yes, it's subjective, but these days even ImageNet is considered small by many).
> >
> > Let me explain: The main question I'm asking myself (and you) is whether this improvement by augmenting through time will remain if you apply your approach to a larger, more diverse (video) dataset. What's the problem with trying that? Fitting ImageNet-scale models can be done these days on a consumer GPU in a few days.
> >
> > Regarding your argument that you're using the exact set of images for both approaches: Doesn't that mean that in plain SimCLR you're using images as negatives that are highly similar, because they're adjacent frames from a video (the positives from -TT)? Doesn't that artificially hurt your SimCLR baseline, in comparison with a more typical use case for SimCLR where the images in the dataset are unrelated and not sourced from videos?
> >
> > Basically my problem with the paper is your claim "that time-based augmentations during natural interactions with objects can substantially improve contrastive learning" (abstract), for which I don't see evidence in the paper. You present evidence that instead of using contrastive learning on images that have been sampled from a relatively small set of videos one should use time-based augmentations. I would support this claim, but it's a substantially different one from the one you make, because few people who uses contrastive learning use it in such a way. The implication of your claim, however, is that instead of doing contrastive learning on image datasets, one should take videos and use time-based augmentations, but you have not actually shown that this approach actually helps when the baseline SimCLR has access to i.i.d. images instead of highly correlated ones.

---

> > > ### Author Response · Authors · 2022-11-18
> > > **Dataset size and negative pairs**
> > >
> > > Regarding your point about data set size: The datasets we’ve chosen allow us to carefully test specific hypotheses regarding the importance of 3-D rotations or seeing the same object in front of different backgrounds for a particular task, which is categorization. We think it is very important to try to understand and demonstrate what kind of video material containing what kind of interactions will have what effect on the representation. We hope you agree with this. Using a giant but uncontrolled video dataset will not help with that. We do not know of any bigger data set that would allow us to systematically address the above questions, but we would be happy to get some pointers from you. Furthermore, we do not see any evidence suggesting that our results shouldn’t hold up for larger data sets. We think the proper thing to do is careful statistical testing using held-out test sets. That’s what we’ve done.
> > >
> > > Thank you for explaining your concern. It seems you suspect that we are artificially crippling SimCLR by systematically taking adjacent frames as negative pairs. This is not the case. Our negative pairs come from images sampled randomly i.i.d. from across the entire dataset (different video clips, different objects, different contexts), just as you rightfully demand.  Because of this, there is only a small probability that negative pairs are constructed from adjacent frames. However, the (small) probability of creating a negative pair from similar images is the same for the different approaches we compare, so it shouldn’t affect the fairness of the comparison. Furthermore, we demonstrate consistent advantages also in the approaches that do not use any negative pairs, like BYOL. Therefore, we think our results support our claims. Please also note that none of the other reviewers criticizes that our claims are not well-supported. Regarding the disputed claim from the abstract, we deliberately use the word “can” to not overstate any implications.
> > > How SimCLR or other self-supervised learning approaches are generally affected by having similar/correlated images in the training set is indeed an interesting question to which we do not know the answer.

---

> > > > ### Comment · Reviewer_Qi6u · 2022-12-07
> > > > **Good point**
> > > >
> > > > I take your point about BYOL not needing negative pairs. I missed that indeed.
> > > >
> > > > However, I do stand by my point that the datasets are relatively small and in many cases the samples are highly correlated (e.g. Core50 has only 50 objects in the videos), which is not a great testbed in my opinion.
> > > >
> > > > Also, I don't see many insights gained in terms of what kind of video material containing what kind of interactions will have what effect on the representation.
> > > >
> > > > In any case, as I seem to be the only one who is not so happy with this paper, I'm giving it the benefit of the doubt and increase my score.
> > > >
> > > > Based on the discussions, I think Reviewer S6Vj should do that, too, since their argument was purely semantic and the authors' response does address it well in my opinion.

---

### Official Review · Reviewer_fkWg · 2022-10-25

**Confidence:** 5
**Correctness:** 4
**Technical Novelty And Significance:** 3
**Empirical Novelty And Significance:** 3
**Recommendation:** 8

**Clarity, Quality, Novelty And Reproducibility:**

The paper is easy to read, very clear, and the appendix and level of detail is high enough to guarantee reproducibility. Authors also have states in the paper that they will share their code if the paper is accepted and that is a great idea.

**Strength And Weaknesses:**

See Summary of Review in section below. TLDR: The main paper's strengths are the original scientific question, and methodological execution (dataset, networks and learning regimes used). The main weakness is a tentative missing control to justify the main claim (see summary review section for details).

**Summary Of The Paper:**

This paper proposes the idea of using time-based data augmentations to aid general purpose machine vision systems. Authors provide a thorough and very detailed psychological motivation for why this may work, and later use simulated agents's view points to simulate the time-based components of learning. Authors find that stacking time-based representations on top of other data-augmentation schemes aids in recognition.

**Summary Of The Review:**

This is a type of paper that everyone talks about that "someone will have to do this", but no one is aware of anyone doing it, so I am happy that a paper regarding training a NN that incorporated the temporal dynamics in a SSL framework is studied and properly evaluated. This paper deserves an accept just because: 1) the authors have tried (and this isn't an obvious thing to do both in terms of finding an interesting scientific question and also the methodology, since most of the CV community is unfortunately still thinking about ImageNet, and developing a convoluted new architecture to increase the validation set performance by 0.1%); 2) authors find a set of interesting results and the cohesive story of the experimental design + evaluations mostly make a lot of senses (however, there's a couple of questions I have later on this).

The main **weakness** I find in this paper is that I think at some point there was some conversation (in the paper) about having a control condition where the video frames were shuffled as a control to eliminate the confounding variable of "more data". Without this control, the paper almost falls short because it still raises the question: **"Do time-based augmentations drive this increase in performance due to their time-based nature, or by pure virtue of adding more training data?"** Maybe authors have added these plots in the Appendix and I've missed this?

Other than that, **this is a fantastic paper(!). I'm willing to increase my score if authors address my concerns in this review.**

* Missing References:

   * On eye-movements and multiple views for robustness, it may be worth adding the recent work of **Harrington & Deza. ICLR 2022** that showed through a set of psychophysical experiments that learning of multi-view foveal and peripheral template potentially aids in robust representations for humans (through adversarial robustness experiments in machines as well).

   * Another key paper that is missing is the theme of time-based contrastive/self-supervised models are: ``Are models trained on temporally-continuous data streams more adversarially robust?'' by **Kong & Norcia. SVRHM 2021.**

Of course, it would be quite interesting to see how time-based learning affects the agents performance in adversarial robustness and/or common corruptions. Perhaps this is not doable to do for the rebuttal (that is ok), but I'd be curious what the authors would think would happen. I expect stronger robustness, but will this actually be the case if some studies have shown that a robustness-accuracy tradeoff exists? (Tsipiras, ICLR 2019)

---

> ### Author Response · Authors · 2022-11-16
> **Response to review**
>
>
> - Do time-based augmentations drive this increase in performance due to their time-based nature, or by pure virtue of adding more training data?
>
> We would like to clarify that it is indeed the time-based nature of augmentations that drives the performance increase and not an increase in training data. Indeed, our baselines without the temporal augmentations have access to the same set of source images and thus use the same number of training inputs. We also control for the number of positive pairs.  We have added Figure 3 to visualize how we augment our inputs. We have also reformulated Section 3.5 to clarify this.
>
> - Can time-based augmentation improves performance with respect to adversarial robustness and/or common corruptions ?
>
> Thank you for the interesting discussion. It looks like an exciting direction of work. We speculate that time-based learning will aid with adversarial robustness, because it emphasizes 3-D shape features in the representation rather than texture features.

---

> > ### Comment · Reviewer_fkWg · 2022-12-08
> > **Increasing Score, Thanks**
> >
> > Dear Authors,
> >
> > Thanks for clarifying this doubt (on the extra experiment + dissociation). I think this paper would make a great addition for ICLR as the general topic of how time-based representations can help with learning has been popping up in many talks and academic conferences, so this paper would be quite timely!

---

### Official Review · Reviewer_UnyN · 2022-10-28

**Confidence:** 4
**Correctness:** 4
**Technical Novelty And Significance:** 3
**Empirical Novelty And Significance:** 3
**Recommendation:** 8

**Clarity, Quality, Novelty And Reproducibility:**

Clarity: Due to the complexity of the experiments, it is not always easy to follow what was being done.

Quality: This is a very thorough analysis of the effects of "natural" interactions on object recognition. The results are impressive.

Novelty: While temporal augmentation has been done before, this paper investigates it in a very thorough and systematic manner.

Reproducibility: the code will be made available.

I didn't see any figures showing what the stimuli look like for 3DShapeE? Did I miss it?

p2: have shown -> have been shown
allowing to discard _> allowing the model to discard
proposed to use -> proposed using
proposes to learn -> proposes learning

p5: we approximate different kind of -> we approximate different kinds of

p6:, near top: replace the neither nor construction with either or.

p7: 2nd line from bottom: It means that -> This means that

I'm unclear on what the "speed of object rotations" means in terms of the sequence of stimuli. Does this mean that object pairs may be in completely different orientations? A picture would help a lot here. It could be in the appendix, if necessary.

p8: allows to encode -> allows the model to encode

encoding the shape similarity makes easier category ->
encoding the shape similarity enables easier category

similarity makes easier -> similarity enables easier

p9: embodied agents that interacts -> embodied agents that interact


**Strength And Weaknesses:**

Strengths:

+ The idea is well-motivated by data from infants
+ The improvements are substantial, and impact three different categories of self-supervised systems.
+ The analysis of the different factors affecting the outcome is fairly good.
+ The method is tested in simulation, where the kinds of interaction can be controlled, and on real-world video of object manipulations.

Weaknesses, with concrete, actionable feedback

- The idea is not completely novel; several authors have proposed similar ideas. The difference here is the thoroughness of testing, the systematic analysis, and the very large datasets used, generated via simulation.

- The paper is somewhat dense, due to all the experiments conducted, and is a little difficult to follow at times.



**Summary Of The Paper:**

This paper proposes and experiments with the idea of time-based augmentations: Human infants manipulate and move objects in front of themselves, move their eyes, making multiple fixations on an object, and carry objects into different rooms, providing independence of object appearance from background. All of these manipulations can be thought of as time-based augmentations of the data. Using these ideas with self-supervised learning provides a biologically-inspired way of learning representations of objects. The paper tests these ideas on three different self-supervised algorithms on four different datasets, and finds that adding time-based augmentations to standard ones significantly increases performance. The improvements are not in the 1-2% range that some papers show; rather, many results show a 10-20% increase, suggesting a real impact on self-supervised learning. Interestingly, time-based augmentations are not by themselves particularly effective (except in simulation); it is only in combination with standard augmentations that there is a substantial effect.

I have read the other reviews (which I think are substantially unfair, and/or are from reviewers who, for some reason I fail to understand, want to kill the paper), the authors' rebuttals, and skimmed the updated paper.

My evaluation still stands: This is an excellent and exciting paper, well worthy of publication. It could be a good talk at the conference.

**Summary Of The Review:**

This is an exciting paper, showing that "natural" temporal interactions with objects significantly boosts the performance of self-supervised algorithms.

---

### Decision · Program_Chairs · 2023-01-20

**Decision:**

Accept: poster

**Justification For Why Not Higher Score:**

The idea is not completely novel; several authors have proposed similar ideas.

**Justification For Why Not Lower Score:**

There is unanimous agreement that the paper should be accepted.

**Metareview: Summary, Strengths And Weaknesses:**

This paper describes a biologically-motivated approach for time-based data augmentation to be used for self-supervised learning. There was general agreement that the reported improvements over baseline and that the approach was interesting at least from a biological perspective

**Note From Pc:**

if the above contains the word "oral" or "spotlight" please see: "oral" presentation means -> notable-top-5% and "spotlight" means -> notable-top-25%. As stated in our emails, we are disassociating presentation type from AC recommendations